# Strengthening Sugar-Sweetened Beverage Taxation for Non-Communicable Disease Prevention: A Comparative Political Economy Analysis Case Study of Fiji and Tonga

**DOI:** 10.3390/nu14061212

**Published:** 2022-03-12

**Authors:** Sarah Mounsey, Aspasia Katrina Vaka, Tilema Cama, Gade Waqa, Briar McKenzie, Anne Marie Thow

**Affiliations:** 1Menzies Centre for Health Policy and Economics, School of Public Health, Charles Perkins Centre, University of Sydney, Sydney, NSW 2006, Australia; annemarie.thow@sydney.edu.au; 2Queen Salote Institute of Nursing and Allied Health, Ministry of Health, Vaiola Hospital, Nuku’alofa, Tonga; linavaka@gmail.com (A.K.V.); tilemacama@yahoo.co.nz (T.C.); 3Pacific Research Centre for the Prevention of Obesity and Non-Communicable Diseases (C-POND), Fiji National University, Tamavua Campus, Suva, Fiji; gade.waqa@fnu.ac.fj; 4The George Institute for Global Health, The University of New South Wales, Sydney, NSW 2042, Australia; bmckenzie@georgeinstitute.org.au

**Keywords:** sugar-sweetened beverages (SSBs), non-communicable disease (NCD) prevention, diet-related fiscal policies, political economy analysis, policy analysis

## Abstract

Diet-related fiscal policy is an effective NCD prevention strategy. However, current sugar-sweetened beverage (SSB) taxes in Fiji and Tonga have not had the desired effect; SSB consumption in Fiji, for example, contributes to mortality more than double the global rates and is highest in the WHO Western Pacific Region. We therefore aimed to better understand the perceived underlying political economy drivers that have and continue to affect change in each country. Our study design utilised a comparative case study that triangulated documentary policy and stakeholder analysis with semi-structured stakeholder interviews in both countries and an in-depth corporate political activity analysis in Fiji. We drew on theoretical frameworks relevant to political economy to collect and analyse policy and stakeholder data, and utilised established corporate political activity frameworks to analyse industry activity. Common findings to both Fiji and Tonga suggested that the SSB tax impact could be increased through multisectoral engagement, embracing a whole-of-society approach, strengthened institutional structures and leveraging off competing priorities across sectors towards more common goals. These findings provide opportunities and lessons for Fiji and Tonga as well as other similar settings seeking to strengthen or upscale the impact of diet-related fiscal policy.

## 1. Introduction

In 2020, the World Health Organization (WHO) reported non-communicable diseases (NCDs) were responsible for 71% of global deaths (41 million of the 57 million deaths) [1]. Globally, the metabolic NCD risk factors of high systolic blood pressure (SBP), high fasting plasma glucose, high body mass index (BMI) and high low-density lipoprotein (LDL) cholesterol, alongside the joint effects of 15 dietary behaviour risk factors, contributed to the top ten global attributable deaths and disability-adjusted life years (DALYs) in both males and females across all ages [2]. The DALYs alone translated to nearly 8 million deaths globally in 2019 from dietary risk factors in adults aged 25 and over, with a further 30 million deaths attributable to the metabolic risk factors. Additionally, global mortality from just the consumption of high sugar-sweetened beverages (SSBs) was estimated to contribute to 242,000 deaths globally (95% CI 172,000–302,000) [2].

In the small island Pacific nations of Fiji and Tonga, with populations of approximately 900,000 and 105,000, respectively [3], NCDs were responsible for 84% and 83% of total mortality. Global Burden of Disease (GBD) data indicated that in 2019, Fiji’s mortality rates from high SBP, fasting plasma glucose, high BMI and high LDL cholesterol were all nearly double those of Tonga and nearly triple the global rates (for high fasting plasma glucose and high BMI) [2]. In Fiji and Tonga (as with the rest of the Pacific region), SSB consumption is significant [2]. Nearly one percent of total mortality in Fiji was attributable to SSB consumption—double that seen in Tonga—however, both countries are above the global average of 0.4 percent [2].

The United Nations High-level Political Declaration on NCDs in 2011 [4] initiated the development and implementation of the Global Action Plan for the Prevention and Control of Noncommunicable Diseases 2013–2020 [5]. This declaration informed the development of a roadmap and a set of policy options for member states to reach nine voluntary targets aimed at reducing exposure to NCD risk factors through the development of multisectoral NCD plans and strategies. The reduction in NCD-related premature mortality is also included as a specific target of the Sustainable Development Goals (SDGs) (SDG goal 3.4, “by 2030, reduce by one third premature mortality from NCDs”) [6]. Reducing sugar consumption has been identified as a priority, globally, due to its contribution to diet-related NCDs [7,8]. One of the key interventions identified to reduce sugar consumption has been SSB taxes, considered a WHO “best-buy” NCD policy intervention [9,10]. However, implementation of effective interventions to address sugar consumption has proved challenging globally, in the face of strong industry resistance and cross-sectoral policy tensions [11,12,13,14].

Despite high level government commitment to address what is termed the “NCD crisis” in both Fiji and Tonga, recent evidence indicates that malnutrition and NCD burden continues to be an issue of concern [15]. Although Fiji and Tonga have both implemented a range of policies for the prevention of diet-related NCDs, including fiscal policies, they have not always had the desired impact, and there is recognized scope to strengthen both policy design and implementation [15,16]. In 2018, Fiji increased the import excise duty for SSBs to FJD 2.00/litre (approximately USD 0.91) with a separate tax for locally produced SSBs of FJD 0.35/litre (around USD 0.17) [17]. In 2013, Tonga first introduced a 1 Pa’anga/litre (around USD 0.05) excise tax on soft drinks containing sugar or sweeteners, as well as various excise taxes on imported mutton and lamb flaps, turkey tails, chicken leg quarters, lard, dripping and mayonnaise [18].

Our research sets out to further understand the strengths of and challenges to effective SSB taxation in Fiji and Tonga, as a best practice nutrition policy intervention that requires cross-sectoral policy engagement. In particular, we aimed to (1) identify the nature of relevant state and non-state actor perceptions and concerns regarding the broader policy context for NCDs and, specifically, diet-related fiscal policy, and (2) to identify influences on nutrition policy more broadly but with special focus on diet-related fiscal policy design, decision making and implementation (including politico-economic, cultural and process factors). Our study was innovatively grounded in political economy analysis (PEA), which is an increasingly utilized and powerful tool to provide a detailed, nuanced understanding of the power, state capability, accountability and responsiveness of the stakeholders and donors involved in a policy cycle [19,20]. For food and nutrition-related fiscal policy, a systematic PEA approach can help to identify the relationships and interactions, interests, incentives, cultural and political constraints to, or opportunities for, shaping and strengthening policy implementation and accelerating progress for improved nutrition [20].

## 2. Research Methodology

### 2.1. Study Design

Our study design drew on PEA and policy analysis methods as a comparative case study. The study design was also informed by case study research methods, which are recommended for analysing a naturally occurring phenomena over which the researcher has little or no control; in this study, the increased diet-related fiscal policy [21,22]. We combined data from a documentary policy analysis, stakeholder analysis and stakeholder interviews, triangulating data in the analysis [23].

### 2.2. Study Theoretical Framework

The use of political science frameworks has been recommended for analysing the political economy of nutrition policy [24]. Two theories relevant to political economy, which focused on policy agendas and priority setting, underpinned the data collection and analysis. Together, these theories highlight the importance of actors and institutions; frames and ideas (particularly with respect to the nature of the policy problem and solutions); interests and power; context; resources, and support the examination of core political economy considerations. First, according to Shiffman’s framework for global health policy priority setting, the political context (or the environments in which actors operate), provides the governance structure (or the degree to which political actors operating in a sector provide the opportunity for collective, coherent action) [25]. Second, Kingdon’s “multiple streams” theory argues that for an issue to gain attention on a government’s agenda and policy change to occur, there must be an intersection in understandings and actions regarding the policy problem, the proposed policy solutions to the problem and political events [26].

### 2.3. Data Collection

First, we used a policy documentary analysis approach to review key policy documents across sectors relevant to food and nutrition in Fiji, (*n* = 11) and Tonga (*n* = 7) (Appendix A and Appendix B), to identify current priorities and activities relevant to diet-related NCD prevention and the application of fiscal policy specifically in this context. Inclusion criteria were policies that were, to our knowledge, current at the time of our research and relevant to diet-related NCD prevention. Policies were identified through internet searches and direct requests to government officials from relevant sectors. We developed a matrix in Microsoft Excel to collect relevant information extracted from the relevant policies, including the policy objectives, responsible ministry, and activities and priorities relevant to diet-related NCDs. The use of such a matrix is useful to identify the key content of policies relating to NCD prevention: multisectoral engagement, framing and beliefs of NCDs, the resourcing and accountability of policy actions, gender considerations and implementation/enforcement mechanisms. Through the use of the detailed matrix, we were able to immediately identify gaps in policy content that could then be focused on in more detail.

The second source of data was from interviews (Fiji *n* = 18 and Tonga *n* = 24), conducted by the lead researchers and in-country colleagues with policymakers and stakeholders from within the relevant government sectors (health, finance, agriculture, industry, employment etc.), private sector and external actors who could contribute to a policy decision via cross-sectoral “policy networks” [27]. Inclusion criteria were that the participants had an interest or influence in or were impacted by the diet-related issue. The participants were from comparable sectors across both countries and relevant to the issue of SSB taxation. Interview respondents were identified and recruited through our in-country colleagues via email, in-person visits and telephone calls. The interviews were recorded, semi-structured, key-informant interviews conducted virtually (Fiji) and in person in Tonga (Nuku’alofa). Each lasted approximately 60 min. We conducted interviews in Tonga November–December 2019 and Fiji January–May 2020. The interviews centred around respondents’ perceptions and understanding of diet-related NCDs as a policy “problem” and fiscal policy—as well as other salient interventions—as policy “solutions”. Specifically, this component of the interviews focused on governance, resources, perceived barriers and potential opportunities to address NCDs. Interviews were transcribed in full using NVivo transcription software. Validation of the data involved providing all researchers a copy of the codes and discussing any discrepancies for the deductive codes.

### 2.4. Data Analysis

We used a usual approach for qualitative, comparative case study research that comprised of two key components. First, we analysed each country separately and then conducted the comparative analysis [28,29].

We analysed the policy documentary data by sector and by content themes to identify the nature of the policy content in each case study country. Next, we determined relevant state and non-state actor perceptions and concerns regarding NCDs and the implemented diet-related fiscal policy. Within our framework, pre-determined themes included: specific “best practice” policy content for diet-related NCD prevention (based on international recommendations); framing and beliefs evident regarding the “policy problem” of NCDs; resources; governance; gender; implementation. We also used thematic analysis to identify influences on policy design, decision making and implementation of the diet-related fiscal policy. We then integrated this data with NVivo software to code the transcribed interviews based on the elements of the theoretical framework. Following this, we analysed the coded data to identify key themes in response to each of the study aims until theoretical saturation was achieved [21]. There was also an element of iterative analysis; the documentary data informed the interviews by providing specifics on the nature of the case study as the basis for questions on the agenda setting and policy processes.

Validation of this component involved circulation of the matrix to all researchers and in-country colleagues, followed by a discussion of any discrepancies in the findings or suggestions for inclusions or amendments.

## 3. Results

Our analysis highlighted a number of future opportunities for strengthened diet-related fiscal policy for NCD prevention in both Fiji and Tonga. Key findings from both countries indicated the potential for increased multisectoral engagement and to embrace a whole-of-society approach, to strengthen enforcement, monitoring and evaluation mechanisms, and to leverage off consistent competing priorities and policy objectives across sectors for more inclusive, nutrition-sensitive actions. We found Fiji had specific challenges around the influence of industry and the overall economically-focused strategic direction of government. Here, we present in-depth country-specific findings for Fiji and Tonga followed by comparative findings between countries. Appendix A and Appendix B provide a list of included policies, years active and objectives relating to diet-related NCDs.

### 3.1. Fiji Specific Findings

Our analysis of Fiji’s interview data revealed diet-related fiscal policy was considered “low-hanging fruit” for reducing the consumption of unhealthy products. However, the majority of respondents indicated that taxed products’ cost increases did not really impact people’s purchasing behaviour and, overall, taxes were ineffective. Underlying drivers most frequently noted from both documentary analysis and interviews were: (1) competing policy objectives and priorities; (2) inadequate multisectoral engagement and collaboration; (3) insufficient enforcement, monitoring and evaluation; (4) actor influence. Mediating factors influencing dietary behaviour were largely perceived to be changing working patterns and busy lifestyles where time constraints strongly influenced convenience, particularly in the more populated urban areas, as well as a lack of nutrition knowledge and awareness at the community level. We found respondents across all sectors acknowledged that complementary processes to strengthen the tax policy would involve incentive, as well as an inclusive, bottom-up approach from government in the consultation, communication and decision making of implemented taxes. We now look at each underlying driver in turn.

#### 3.1.1. Competing Policy Objectives and Priorities

Our analysis found all but one respondent perceived competing policy objectives and priorities were a major barrier to reducing NCD burden. The long-term nature of NCD prevention policies means implementation is often dominated by a political agenda either of an acute nature (e.g., COVID, cyclones) or where results would align with a defined political time horizon [30]. From our documentary and interview data we found two competing priorities were economic development and climate change. Many respondents indicated that even when an increased SSB tax was recommended to the government, any interest was more aligned to revenue generation than it was to improved health outcomes. Respondents commented:

“…I think you always have competing challenges…trying to get nutrition anywhere near the top of the agenda is just so difficult because there’s always something that is distracting…”Civil Society

“…I just feel that we are so economically focused, we want more money. They are saying ‘wealth is health’…”Policy Advocate

With economic growth firmly the number one priority in Fiji, policy tensions were clearly a challenge with respect to NCD burden. Many respondents noted policy focus was more on improving infrastructure and productivity for commercial agriculture and expanding the export market than on improving nutrition security for the largely rural population and the domestic market. We also noted an element of “now is not the time for taxes on food” given the economic impact of COVID-19 and its shaping of food security.

Natural disasters pose a genuine risk for Fijian communities. For instance, tropical cyclones Winston (2016) and Gita (2018) devastated crops and destroyed whole communities. Thus, policy priority for disaster preparedness and response and tackling climate change will always be critical, but providing nutrition-sensitive policies, as suggested by three respondents, that supported healthier dietary behaviours (such as an SSB tax), would allow public health improvements over time.

Subsequent improvement to employment productivity was raised by two respondents who suggested presenteeism (being present at work but sick and less productive) was a key problem in Fiji and was counterproductive to the government’s strategic direction for an improved economy. Therefore, tackling health and NCDs underpinned improved productivity.

“…I personally feel there needs to be focus on nutrition and health because proper nutrition contributes to proper health and wellbeing. It will reduce the cost of our healthcare in Fiji, it will make our economy more buoyant, more successful…”Civil Society

#### 3.1.2. Limited Multisectoral Engagement and Collaboration

Our analysis showed the next most common opportunity perceived by respondents was adopting a whole-of-government, whole-of-society approach for improved policy response. Our document analysis indicated a platform of multisectoral policies; however, collaborative action across sectors was declining. Moreover, many respondents felt that it was not just high-level ministries but all stakeholders that needed more engagement:

“…You need intersectoral collaboration. It’s so important because health alone cannot do this. But there’s been limited progress and the different sectors are still kind of working in isolation…”Civil Society

#### 3.1.3. Lack of Enforcement, Monitoring and Evaluation

Respondents perceived that there was limited enforcement and monitoring and evaluation (M&E) in the policy process, including for SSB taxation, which linked to the level of political support and resources. Respondents identified this as resulting in a lack of evidence regarding severity of the problem and the (potential) effectiveness of specific policies. However, there were respondents who indicated substantial efforts were being made:

“…It’s not only important to make policy on unhealthy products but I think the important thing is how these are really followed. There’s a gap in that—no monitoring, no evaluation of the policy. There is a lot of strengthening required…”Academia

#### 3.1.4. Actor Influence

Our documentary analysis found government demonstrated commitment and were taking action on NCDs, and that public interest objectives were clearly evident in policy. However, we found interview respondents repeatedly suggested that due to the political context and overall strategic direction of Fiji to achieve greater economic growth—particularly in respect to accessing the global market—the larger food and beverage manufacturers wielded the most influence. This influence, respondents argued, was underpinned by the drive for profit, not health, and, thus, strategies to shape government decisions directly (or indirectly) targeted employment, revenue generation and economic growth. Respondents commented:

“…Industries, like corporations. They’re the ones who have more influence. We’re a government driven by the need to gain money in the shortest time frame. That’s our priority and everything else falls within that priority…” Civil Society

We found that while respondents acknowledged this influence, they also recognised the necessity for industry to be involved in the NCD policy process and thus a significant part of the solution. Indeed, the three interviewees from industry showed genuine engagement with government, particularly if government were prepared to subsidise costs for the resources required to drive change; policy champions and academics indicated industry regularly attended seminars and high-level meetings and were actively interested in efforts to address the NCD burden.

“…I think it’s really important to continue to engage with the private sector. The government develops policies but it’s actually the private sector who plays a very important role in terms of food production and in the distribution and retailing of the [less healthy] food products that are there in the market. So, it’s really, really important, I think, to engage them…” Development Partner

We also found respondents perceived an opportunity to tap into the influence of specific actors, to promote stronger SSB taxation from the “bottom up”. In particular, community voices, policy champions, international organizations, civil society, academia, etc., whose advocacy and global expertise could help drive the policy issue forward. In fact, a central theme was that of ensuring constituencies and communities were heard in the policy-making process, thereby fostering an inclusive approach in which the needs from all levels of the Fijian population were considered—and met—and no one was left behind. Additionally, participants perceived that government would be more likely to successfully implement policy actions when the public were involved in order to establish future support. We illustrate:

“…One approach is to move more into a community approach to those who have interests in nutrition. That’s how I try and mobilize the community—to inform communities better in the hope that they will be able to come together and ask governments to change policies on NCDs, or to make those local policies. It’s a bottom-up approach, as I’m finding a lot of resistance and hitting the wall at government-to-government level…”Government

#### 3.1.5. Opportunities for Strengthening Diet-Related/NCD Policy

We found that SSB taxation, and indeed diet-related fiscal policy broadly, was not high on the political agenda, in terms of a clear objective to strengthen taxation for better health outcomes. However, four respondents indicated that the policy community is continuing to refine policy solutions, identify linkages with what is currently the government’s strategic direction (i.e., economic development, climate change, disaster preparedness) to “couple” policy actions and provide evidence for their arguments.

### 3.2. Tonga Specific Findings

In Tonga, analysis of the 24 interviews indicated that the taxes, while accepted and welcomed by all but two (industry) respondents, were perceived by most to be less effective than expected for reducing consumption of the taxed products—respondents still purchased the unhealthy products and indicated they would just save more money to buy them. Underlying mechanisms driving this related to: (1) framing—many respondents felt complementary consultations coinciding with the implementation of the taxes were insufficient to help them understand why the policy was imposed; (2) the fragmentation of the policy process between government ministries; (3) inadequate healthier substitution products. Mediating factors that affected dietary behaviours included culture and tradition. However, these were also considered opportunities for strengthening the tax policy with complementary policy actions: education, improved infrastructure and the influence of powerful actors figured highly. We describe each finding in turn.

#### 3.2.1. Framing

We found that respondents were unanimous that NCDs were the main health issue facing Tonga, and the obesogenic environment exacerbated behaviours that lead to the high prevalence of NCD burden. They also indicated minimal health literacy and awareness, the convenience and low price of unhealthy, imported products and an increasingly urban lifestyle as drivers. For example:

“…I think for us, it would be much better if we didn’t have all the other food that we Tongans just go for, that cheap food that they get from anywhere, and then the tendency is to buy and eat to fill the stomach, but not necessarily understanding what they are eating…”Church

We found all respondents felt taxes were not discouraging people from consuming unhealthy products. This was confirmed by a respondent from the sales sector who indicated there had been no change in the volume of ice cream and SSBs purchased. More importantly, respondents suggested the tax would hurt low-income households and threaten vulnerable populations.

“…So, I believe that we do need these taxes to some degree only, because it may have huge negative impacts on the livelihoods of the poor and the vulnerable population. We need to be very tactful and very wise in our taxation approach because the totality of impact may be negative at the end, to their health, which was the purpose at the beginning [to improve health]…”Government

#### 3.2.2. Fragmentation of the Policy Process

Our analysis identified a fragmented policy process. Respondents generally understood how the tax was developed and acknowledged multisectoral collaboration; however, many were unclear of who initiated the taxes and what products were included. All respondents recognised different government interests driving the tax as well as differing roles and mandates regarding the policy design and implementation.

“…I think there’s no mechanism to link them all together. I think it all depends on the strategies and the vision of every Ministry…”Government

#### 3.2.3. Lack of Healthier Substitution Products

From our interviews, respondents suggested healthier alternatives to the taxed products were minimal. Over half the respondents felt that without the option for alternative, inexpensive foods to replace the taxed ones, the policy just was not feasible and, furthermore, would potentially drive low-income households into poverty.

“…For the noodles and the imported food, we have to be careful—we have to give alternatives. If we are encouraging women to use healthy ingredients, they have to be available. If they don’t grow it, it has to be available in a marketplace and be affordable…”Church

### 3.3. Comparative Findings

Our documentary and interview analysis showed Fiji and Tonga shared similar challenges and opportunities relating to the implementation and success of diet-related fiscal policy. Here, we describe four common themes emerging. In both countries, there were cultural and societal factors that were perceived as mediating the impact of SSB taxes in contributing to SSB taxation. Table 1 provides some insight into respondents’ perceptions around the most common findings for our case study countries. These relate to socio-cultural factors including an increasing migration away from rural living, health awareness, culture and tradition, busier working patterns and food security.

Second, natural disasters and climate change figured highly. Respondents perceived their vulnerable geographical location was a significant barrier for NCD-related policy objectives to be achieved—particularly for reducing imported, less healthy products. Cyclones and floods are an annual threat to the whole Pacific region and can be a significant setback to each countries’ socioeconomic development, irrespective of ensuring sustainable food security. For instance, in Tonga, Cyclone Ian (2014) resulted in an 11% drop in coconut imports, 98% of the root crops were destroyed and pumpkin and gourd dropped by 87% (2016 value) [31].

Third, we noted the key influence of high-level actors in each country that can act as policy “champions” to advocate and push policy objectives. Our analysis found this untapped influence was perceived to be a drawback for both Fiji and Tonga. For both countries, political will is strong, with clear objectives for NCD prevention evident in Fiji’s National Development Plan and TongaHealth’s NCD Strategic Plan. However, participants perceived more could be achieved through engaging other actors with influence and power to drive these policies forward. One Fijian respondent commented:

“…I know for Fiji the president is a real champion for food security and NCDs but he hasn’t been utilized very well. We haven’t engaged him as much, but I know he’s willing to support it, to put his weight behind these issues. If we can engage him and support him as a champion for food security and NCDs, I think he’d be willing to get the government on board. But I haven’t seen the Ministry of Health engaging him more…”Development Partner

In Tonga, four respondents indicated the King and Queen of Tonga showed strong commitment towards a “Healthy Tonga” and described “health” as one of the key priorities for government action. Moreover, all respondents perceived churches as having ultimate influencing power to enable behaviour change. We illustrate:

“…The King, he came to an opening of parliament appealing to the people how important it is to be healthy and peaceful. To me, if I am in charge of a sector, I will go out of my way to try to coordinate the sectors, have a national committee or something to follow what the King has expressed…”Government

“…I know if our church decides to do away with these things [unhealthy products] then that can have a big impact and authority…”Civil Society

Finally, and linking closely with high-level actor influence, is the institutional structure and the political landscape. We found that in Fiji and Tonga, decisions regarding increases or changes to the taxes were top-down, making it difficult for groups external to government to influence the decision-making process. The literature on political economy analysis suggests that in contexts with less robust “formal” institutions (where normally, constitutional rules and laws would regulate power and influence) more “informal” institutions (where political, socioeconomic and cultural norms determine how an issue is actually played out) may create tension between the policy process and incentives for change [19]. Our theoretical framework also suggests that changing the institutional landscape can affect policy action and process. Interestingly, our findings indicate this can work both ways.

In Fiji, for instance, the lapse of the formal institutional structure that once encompassed a multisectoral task force to focus on nutrition and NCD prevention—including diet-related fiscal policy—has meant no further scale-up of fiscal policy, either in the form of increased taxation or enforcement of taxation already in place. Many respondents claimed taxation made no difference, should be increased and that subsidies were not passed through to consumers. However, the more informal institutional structures inclusive of policy champions, NCD advocates and community groups united for increased SSB taxation at multiple levels, resulting in eventual and unavoidable government action to increase the diet-related taxes.

In Tonga, however, many respondents indicated that taxes could benefit the Tongan population. However, it is the formal NCD steering committee and strong international donor support and activities targeting multisector engagement and commitment over the use and development of health taxes more broadly that has affected change. There is no informal institutional structure advocating for health taxes, and other respondents indicated taxes should be higher as people were still purchasing the taxed goods. These findings emphasize the necessity for political will, strong institutional governance structures and a whole-of-government commitment to underpin and drive policies targeting reduced NCD burden.

Nevertheless, we noted clear differences in challenges to policy impact. We found these pertained largely to the size, economics and cultures of the countries. Fiji, for example has a population nearly ten times that of Tonga and remains the hub for economic and manufacturing output in the Pacific region. The large industry presence is in stark contrast to that of Tonga and changes the political economy significantly—hence, the overall strategic direction in Fiji is based on economic growth and development.

## 4. Discussion

Our study has provided important, current contextual insight into the political economy shaping the diet-related fiscal policy space in Fiji and Tonga. A better understanding of the underlying drivers affecting the diet-related fiscal policy process is critical as these countries strive on the one hand to mitigate spiralling NCD burden, and on the other hand to navigate towards greater economic prosperity. For Fiji, we found these factors included: (1) competing policy objectives and priorities, (2) inadequate multisectoral engagement and collaboration, (3) insufficient enforcement, monitoring and evaluation, and (4) actor influence. For Tonga, these factors included: (1) framing, (2) fragmentation of the policy process and (3) insufficient healthier substitution products. Complementary policy processes to strengthen the tax policy also varied. Common to both Fiji and Tonga were respondents’ perceptions that targeted health promotion and education would significantly strengthen efforts for improved health. For Fiji, with a significant food and beverage industry presence, complementary policy actions would involve incentive, as well as an inclusive, bottom-up approach from government in the consultation, communication and monitoring of the implemented taxes. For Tonga, improved infrastructure and the influence of powerful policy champions were considered critical.

While economic considerations were common to both countries, they were particularly evident in Fiji. For instance, our analysis revealed the food and beverage industry plays a pivotal role in providing employment and financial security to many Fijians and contributes significantly to the country’s economy. In fact, two of the global “top ten” transnational food and beverage manufacturing companies—Coca-Cola and Nestle—are based in Fiji. If Nestle Fiji were to follow practices of Nestle Malaysia, where the recently implemented SSB tax was absorbed by the company, efforts to reduce SSB consumption in Fiji would be seriously undermined [32]. Our analysis indicated that the strong informal power of this industry has impacted decision making for scaled-up diet-related taxation through intense, targeted strategies aimed at disincentivising regulatory instruments for taxation reform and further undermined political commitment for health outcomes in favour of economic gain [33]. This observation reflects similar findings not just in Fiji but more broadly around the world. [34,35,36,37]. For instance, recent empirical evidence suggests neoliberal processes and World Trade Organization (WTO) member status in Fiji have shaped and intensified multinational influence through the advertising, sale and consumption of low-quality, imported food and beverage products [38]. Attempts to restrict and regulate this influence have been, to date, thwarted. Additionally, new evidence from studies across seven sub-Saharan African countries on SSB taxation cited tension between industry (considered a key contributor to the economy of all of the case study countries) and its influence versus strong regulatory interventions aimed at reducing consumption of the taxed products [11].

However, alongside these findings that resonate with the global literature, our study also identified an optimism among respondents regarding the potential for industry to also be part of the solution. In particular, the use of reformulation incentives to minimize tax burden and/or the use of targeted subsidies for manufacturing equipment were considered by (industry) respondents to be feasible actions towards reducing tensions and achieving global health targets.

We also found that while the inherent challenges for each of the case study countries provided unique opportunities, they also provided lessons for each other. For instance, as the population of Tonga grows and industry expands, they will be looking to their close neighbours for guidance on how to increase their economy and industry output while also halting the rise of NCDs in line with global targets. Likewise, Fiji can draw from Tonga’s experience in the strengthening of their diet-related fiscal policy, which currently leads in the Pacific region in scope and evidence use through their collaborative efforts with international organizations and donor partners [15].

Our study supports previous findings that a comprehensive policy community or coalition of interested actors can encourage political commitment to address dietary behaviours [24,33,39]. This support must come from a bottom-up as well as a top-down approach—or in other words, a whole-of-society approach. An overwhelming number of respondents in both Fiji and Tonga indicated the need to include all members of society to ensure all needs are considered and common goals for policy objectives are determined and agreed. This is consistent with other studies exploring political commitment for nutrition policies more broadly. For instance, concluding on a review of what drives political commitment for nutrition, the authors specified “strategic actions by cohesive, resourced and strongly led nutrition actor networks” as core [40]. In our study we also found that strong leadership and multisectoral institutional structures were perceived as factors that enabled translation of high-level political will into policy action. There may also be scope in the Pacific for more participatory governance structures to support this translation of political will into action. Involving citizens, households and communities in the policy-making process incentivises government actors to continue achieving policy targets once on the political agenda [41].

Our study provides insight into the importance of the broader policy architecture in supporting and fostering SSB taxation and potentially other NCD prevention policies. For example, for Fiji, an enabling National Development Plan and a stand-alone, non-discriminatory and evidence-based nutrient profiling mechanism provide reference points for SSB taxation. In our study, respondents indicated that policies across sectors that embedded multisectoral goals and targets could also help navigate competing priorities. For example, the goal of increased commercial agriculture in Fiji—and particularly for increased sugar production—to target a wider global export market can also include nutrition-specific actions for improved nutrition and food security. Similarly, ensuring disaster preparedness and emergency response to recurring cyclones need not mean limiting the accessibility, availability and affordability of healthy food options. Nothing exemplifies this more than the early government response to COVID-19 in Fiji. In March 2020, the Ministry of Agriculture equipped the Fijian public with vegetable seedlings and gardening equipment that enabled rural and urban households alike to a grow a supply of fresh vegetables during potential lockdowns when access to markets would be impossible. Implementing smaller strategies like this COVID-19 example is also likely to be more successful in the longer term; the wider literature argues “success breeds success”, or delivering results “stimulates and maintains the interest of politicians and other actors” [41].

Our findings underscore the need for a strong governance and institutional mechanism to connect health and NCD prevention more broadly to the policy architecture described above. Only then can common goals and priorities across sectors be identified. However, in Fiji, health respondents raised concerns about the pace of nutrition policy change or improvement to better achieve health objectives. Despite numerous efforts by relevant sectors and actors who unanimously felt diet-related taxation was low-hanging fruit for NCD prevention and revenue generation, the policy providing the actions and governance to ensure institutional structure and, therefore, accountability by ministries, expired in 2018 and, at the time of this study, its replacement was yet to be endorsed. This delay may be in part due to actor influence and economic disincentives, but interview respondents also perceived it as budgetary and resource constraints and their allocation.

In addition, the COVID-19 pandemic has impacted NCD policy processes globally. Although minimally impacted by the pandemic in terms of cases, the Pacific region has not been without the economic decline due to decreased market activity, subsequent employment loss (particularly informal employment) and lower remittances [42]. The impact on health services is difficult to predict. A WHO survey showed three quarters of member states participating in the survey reported disruption to one or more of the inpatient and outpatient services, with rehabilitation services, hypertension and diabetes and related complications’ management being the top three overall services being disrupted [43]. For Fiji and Tonga, with high levels of pre-existing NCDs (and particularly diabetes and hypertension), this is problematic.

However, similar to industry influence, there was optimism from respondents who suggested an opportunity that COVID-19 has provided for the perfect “policy window” to redefine the nutrition and food security policy and to embed latest best-buy practice recommendations and more recent evidence around improving the country’s dietary behaviours and achieving global health targets. Importantly, this opportunity is in line with current requirements for strengthened health services: strengthened government commitment to prioritizing health can help mitigate the inevitable current disruption to NCD screening, monitoring and primary care services.

Our study presents an in-depth analysis of political and economic factors influencing the implementation of diet-related fiscal policy in Tonga and Fiji. A strength of the study is the inclusion of multiple data sources to map and deliver a balanced narrative. Another strength is the comparative nature of the case study as opposed to a single country case study. However, our study also has limitations. Our documentary analysis relied on publicly available information, that can be incomplete and may not reflect all current policy practices. To overcome this limitation, we triangulated the data with stakeholder interviews and validation with our in-country colleagues. Finally, the views of the respondents are subjective and may not be representative of the wider population.

## 5. Conclusions

Our research has generated several important insights for informing policymakers and public health researchers of some key understandings on what has made SSB and diet-related taxes work (or not) in Fiji and Tonga. Similar to challenges inherent in implementing nutrition and NCD policies around the world, and particularly in low- and middle-income countries, our case study countries face challenges around competing priorities, multisectoral engagement, enforcement, industry influence and limited health literacy. However, and more importantly, our respondents across the two countries perceived that with strong political will and adequate institutional structures, these challenges presented opportunities. The opportunities identified in our paper will be useful for ensuring more effective policy development and implementation in Fiji and Tonga but will also be relevant to other Pacific nations and/or similar contexts as well as more broadly for countries yet to implement SSB taxes.

## Figures and Tables

**Table 1 nutrients-14-01212-t001:** Mediating factors influencing diet-related NCD policy.

Mediating Factor	Quotes from Fiji’s Participants	Quotes from Tonga’s Participants
Increasing diaspora to neighbouring Australia and New Zealand (increased remittances to families that increases disposable household income)	Not noted from interviews	“…We travel, we go overseas, we come back, we want that kind of lifestyle…” “…But, you know, it’s just that when people have more access to disposable income that they would otherwise have had, so there’s a lot more access to money to buy imported goods rather than ‘grow your own’…”
Minimal health literacy	“…I believe that education is one of the factors that can improve the way we eat. But it also depends on the parents as well because we cannot be just educating children and the parents at home are practicing something else…” Industry “…We give recommendations to them, telling them to eat two or three serves of vegetables and fruits. It is like, ’What does ‘serve’ mean? How much should they consume?’ They’re not aware of it because if they’re aware, I’ve seen them changing their perception and changing these diets…” Academia	“…There’s a misconception around what’s healthy food and what’s not. So, people just consume with very little understanding or knowledge on what they’re taking in…” “…Lack of knowledge on what they are consuming…”
Tradition and culture	“…When we were brought up, we were with the impression that whenever there’s a function, you get a sugary beverage. And that’s like Christmas for you. It’s cheaper than water, proper fresh orange juice, oils, fruits and vegetables…” Development Partner	“…We love eating. And we love eating the wrong food…” ” …I think it’s a cultural thing. Tonga is a very stratified society. So, there are certain foods that are reserved for large special occasions. But if you’re seen as having that quite frequently, then it might mean you’re in a different class or better off than others…” “…In our culture if you want more people involved it has to be incorporated with food. When I was young, we were eating more fatty food compared to now, but we walked everywhere. Now, you eat and you get in the vehicle and that’s the problem…” Government
Changing working trends	“…It’s the availability of processed foods. Fijians are now becoming like the Australians of the past, the Americans of the past, where everything is go, go, go, go. People are working longer hours and wives now are no longer the ‘housewives’, they are all part of the mad rat race. So, the easiest way to do is just get processed food and warm it up or go to McDonalds or one of these fast-food outlets, eat, and everyone is satisfied…” “…Two of the important factors I think are convenience and time constraints. Because when I see the Fiji population the majority of them (the urban population), will eat once a day away from their homes, or food prepared by the vendors, and not be very much concerned about the nutritional value of the food. They will be more concerned about the taste because that’s how the food sells in the market…Academia	“…One would be the trends of working patterns, working mums and all that…”
Affordability, availability and accessibility	“…I would say price. Availability. And accessibility. Over here, those salty and sweet foods, those are the cheapest you can find in most shops and people who don’t have a lot of money would obviously go for those…” Development Partner	“…I think the availability of the unhealthy food…” Government “…I think it’s the availability and accessibility that will change diet and most of all is the perception that all this imported food is the more superior in terms of nutrition....”

## Data Availability

The data presented in this study are available on request from the corresponding author.

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
