# Peer review of "Strengthening Sugar-Sweetened Beverage Taxation for Non-Communicable Disease Prevention: A Comparative Political Economy Analysis Case Study of Fiji and Tonga"

_nutrients, 2022, doi:10.3390/nu14061212_

Round 1

Reviewer 1 Report

Thank you for the opportunity to review this well-drafted article. This study provide opportunities and challenges for Fiji and Tonga as well as other similar settings seeking to strengthen or upscale the impact of diet-related fiscal policy. Despite the work strengths, I recommend addressing the following comments.

  1. There are too much words in the introduction. Please simplify your sentences and highlight the innovation of this study.
  2. Please tell more details about using a matrix in Microsoft Excel to collect the relevant information extracted from the relevant policies in 2.3 Data collection.
  3. Please describe the exact data collection duration in the methods.
  4. I would like the authors to address these issues in the discussion of the article. The coronavirus pandemic has a great influence on people’s lifestyle worldwide. This may impair the noncommunicable diseases follow-up and the diagnosis. Moreover, the pandemic may lead to the changes of the diet-related fiscal policy process.
  5. You had better show more details about the basic information of respondents in Fiji and Tonga or make a list to show the comparability of respondents in the two sectors.
  6. I suggest that you add some inclusion and exclusion criteria for respondents and policy documents.
  7. The fifth sentence of third paragraph of Introduction has grammatical error, please revise.
  8. The fourth sentence of third paragraph of Result has grammatical error, please revise.
  9. The second sentence of third paragraph of 3.2.1 has grammatical error, please revise.
  10. The font in 3.2 should not be bold, please revise.
  11. The last sentence of the first paragraph of introduction, “95%UI” should be “95%CI”.
  12. There's an extra parenthesis in 2.4 L7-8, please revise.

Author Response

Response to Reviewer 1 Comments

Point 1: There are too many words in the introduction. Please simplify your sentences and highlight the innovation of this study

Response 1: Thank you for your suggestions to simplify the Introduction and to highlight the innovation of the study. The following changes have been made to the text for improved simplicity:

  1. First paragraph, third sentence as follows:

“The DALYs alone translated to nearly 8 million deaths globally in 2019 from dietary risk factors in adults aged 25 and over; with a further 30 million deaths attributable to the metabolic risk factors.”

  1. Second paragraph, last sentence:

“Nearly one percent of total mortality in Fiji was attributable to SSB consumption – double that seen in Tonga - however, both countries are above the global average of 0.4 percent.”

  1. Third paragraph, second last sentence:

One of the key interventions identified to reduce sugar consumption has been SSB taxes, considered a WHO ‘best-buy’ NCD policy intervention.”

  1. Fourth paragraph, first and second sentences:

“Despite high level government commitment to address what is termed the “NCD crisis” in both Fiji and Tonga, recent evidence indicates malnutrition and NCD burden continues to be an issue of concern [15]. Although Fiji and Tonga have both implemented a range of policies for the prevention of diet-related NCDs, including fiscal policies, they have not always had the desired impact and there is recognized scope to strengthen both policy design and implementation.”

Also, in the last paragraph, third sentence, we have included the innovation of the study, which relates to the increasing utilisation of PEA as an analytical tool, particularly within the Pacific region:

“Our study was innovatively grounded in political economy analysis (PEA), which is an increasingly utilized and powerful tool to provide a detailed, nuanced understanding of the power, state capability, accountability and responsiveness of the stakeholders and donors involved in a policy cycle.”

Point 2: Please tell more details about using a matrix in Microsoft Excel to collect the relevant information extracted from the relevant policies in 2.3 Data collection.

Response 2: Some more information about using the matrix is useful, thank you for the suggestion. We have included more explanation of the use of matrices in the last sentence of the first paragraph in section 2.3, Data collection. The text is:

The use of such a matrix is useful to identify key content of policies relating to NCD prevention: framing and beliefs of NCDs, the resourcing and accountability of policy actions, gender considerations and implementation/enforcement mechanisms. Through the use of the detailed matrix, we were able to immediately gaps in policy content that could then be focused on in more detail.”

Point 3: Please describ e the exact data collection duration in the methods

Response 3: Again, thank you for highlighting the ommission of this important methodological information. The duration for interviews in both Tonga and Fiji have now been included as the fifth sentence in the second paragraph in section 2.3, Data collection. The text reads:

We conducted interviews in Tonga November-December 2019 and Fiji January-May 2020.”

Point 4: I would like the authors to address these issues in the discussion of the article. The COVID pandemic has a great influence on people’s lifestyles worldwide. This may impair the NCDs follow-up and the diagnosis. Moreover, the pandemic may lead to the changes of the diet-related fiscal policy process.

Response 4: Including a section on the relevant impact to health services due to COVID is useful and current. The eighth and ninth paragraph have been amended/added to address this important influence.

“In addition, the COVID-19 pandemic has impacted NCD policy processes globally. Although minimally impacted by the pandemic in terms of cases, the Pacific region has not been without the economic decline due to decreased market activity, subsequent employment loss (particularly informal employment) and lower remittances (Tandon A, Oliveira Cruz V et al. 2021). The impact on health services is difficult to predict. A WHO survey showed three quarters of Member States participating in the survey reported disruption to one or more of the inpatient and outpatient services, with rehabilitation services, hypertension and diabetes and related complications management being the top three overall services being disrupted (World Health Organization 2020). For Fiji and Tonga, with high levels of pre-existing NCDs (and particularly diabetes and hypertension). this is problematic.

However, similar to industry influence, we identified optimism from respondents who suggested an opportunity: the COVID-19 pandemic may create the perfect ‘policy window’ to redefine the nutrition and food security policy to embed latest best-buy practice recommendations and more recent evidence around improving the country’s dietary behaviours and achieving global health targets. Importantly, this opportunity is line with current requirements for strengthened health services: strengthened government commitment to prioritizing health can help mitigate the inevitable current disruption to NCD screening, monitoring and primary care services.”

Point 5: You had better show more details about the basic information of respondents in Fiji and Tonga or make a list to show the comparability of respondents in the two sectors.

Response 5: We appreciate the reviewers’ suggestion and have now clarified as follows:

Section 2.3: “The participants were from comparable sectors across both countries, and relevant to the issue of SSB taxation.

However, we wish to note that apart from achieving appropriate sectoral participation, the details of the interviewees are of limited relevance, as the objective of our study was focussed on understanding policy processes from knowledgable participants rather than representation.

Point 6: I suggest you add some inclusion and exclusion criteria for respondents and policy documents.

Response 6: Inclusion criteria for policy documents and interview participants have been included, as below. Many thanks for highlighting.

“Policy: Inclusion criteria were policies that were, to our knowledge, current at the time of our research and relevant to diet-related NCD prevention.”

“Participants: Inclusion criteria were that the participants had an interest or influence in or were impacted by the diet-related issue.”

Points 7-12: Spelling and grammatical errors to be amended.

Response 7-12: All errors revised and corrected. Thanks so much for being thorough in your review.

Reviewer 2 Report

I consider that it is an interesting job but that it need be improved in its current state. I will break down the fundamental elements of my assessment.

In the Abstract, the authors emphasize that their work is based on the triangulation of documentation. However, despite indicating in section 2.3 “Data collection” that 18 relevant documents were located, no analysis of them is presented. They limit themselves to presenting a list of them as appendix 1. Next, the results of the interviews are presented in section 3, but without previously exposing the script that they follow. That is, an analysis of the content of the documents analyzed must be presented, as well as the script of the interviews carried out.

Likewise, the theoretical framework used should be explained. In section 2.2. it is indicated that “Two theories relevant to political economy, and focussed on policy agendas and priority setting, underpinned the data collection and analysis”. Such an extreme must be clearly shown. In this way, the reflection of such theories in the field work carried out as well as in the analysis of the results must be expressly indicated. It is also strange that sections 3.1 and 3.2 present the specific findings for Fiji and Toga and then break them down.

I do not consider that the methodology followed, nor the theoretical contextualization that is carried out, allow the work to be accepted.

Author Response

Response to Reviewer 2 Comments

Point 1: In the Abstract, the authors emphasise that their work is based on the triangulation of documentation. However, despite indicating in section 2.3 *Data collection'" (hat 18 relavant documents were located, no analysis of them is presantod. They limit themelves to presenting a list of them as appendix 1. Next, the results of the interviews are prasenied in socion 3, bur wihoul previously exposing the script that hey follow That is. an analysis of the content of thedocuments analyzed mus be presentod, as well as the script of the interviews carried out

Response 1: We appreciate the reviewer’s comments and queries. We have now clarified as follows:

  1. The documentary data was used for identifying priorities and activities, as outlined in the methods and now clarified, as follows:

First, we used a policy documentary analysis approach to review key policy documents across sectors relevant to food and nutrition in Fiji, (n=11) and Tonga (n=7) (Appendices 1 and 2), to identify current priorities and activities relevant to diet-related NCD prevention and the application of fiscal policy specifically in this context.”

  1. We have explicitly identified the use of documentary data at specific points in the results, including:

Section 3.1 “Underlying drivers most frequently noted from both documentary analysis and inter-views were…”

Section 3.1.1  “From our documentary and interview data we found two competing priorities were economic development and climate change.”

Section 3.1.2 “Our analysis showed the next most common opportunity perceived by respondents was adopting a whole-of-government, whole-of-society approach for improved policy response. Our document analysis indicated…”

Section 3.1.4 “Our documentary analysis found government demonstrated commitment and were taking action on NCDs, and that public interest objectives were clearly evident in policy. However, we found interview respondents repeatedly suggested that due to the political context and overall strategic direction of Fiji to achieve greater economic growth....”

3.3 “Our documentary and interview analysis showed Fiji and Tonga shared similar challenges and opportunities relating to the implementation and success of diet-related fiscal policy.”

  1. With respect to the interview script, the interviews were semi-structured, so it is appropriate to list topics rather than attach a script. We have now expanded on the list of topics covered to provide the reader with a more detailed understanding, as follows:

“The interviews centered around respondents' perceptions and understanding of diet-related NCDs as a policy ‘problem’ and fiscal policy - as well as other salient interventions - as policy ‘solutions’. Specifically, this component of the interviews focused on governance, resources, perceived barriers and potential opportunities to address NCDs.’

Point 2: Likewise, the theoretical framework used should be explained. In section 2 2, it is indicated that “Two theories relevant to political economy and focussed on policy agendas and priority setting underpinned the date collection and analysis". Such an extreme must be clearly shown. In this way the reflection of such theories in the field work carred out as we as in the analysis of theresulte must be expressly indicated. It is also strange that sections 3. l and 3.2 present the specific findings for Fiji and Tonga and then break them down

Response 2: We appreciate the reviewer’s reflections on the theoretical framework.

  1. We have now clarified our approach in Section 2.2 by explaining the rationale for the selection of these frameworks, as follows:

The use of political science frameworks has been recommended for analysing the political economy of nutrition policy (Balarajan and Reich 2016)

Throughout the results, we have clearly addressed the elements of the political science frameworks, including framing, actor influence, policy priorities, institutional structures and policy processes.

  1. With respect to the structure of the results: Presenting the findings separately is a usual approach for case study research – we have now clarified in the methods that this was the approach taken, as follows:

We used a usual approach for qualitative, comparative case study research that comprised of two key components. First we analysed each country separately and then conducted the comparative analysis [27, 28].”

Round 2

Reviewer 2 Report

The authors have answered the elements highlighted. So, I consider that can be accepted for publication.

Best regards.